# Keratin Profiling by Single-Cell RNA-Sequencing Identifies Human Prostate Stem Cell Lineage Hierarchy and Cancer Stem-Like Cells

**DOI:** 10.3390/ijms22158109

**Published:** 2021-07-28

**Authors:** Wen-Yang Hu, Dan-Ping Hu, Lishi Xie, Larisa Nonn, Ranli Lu, Michael Abern, Toshihiro Shioda, Gail S. Prins

**Affiliations:** 1Department of Urology, University of Illinois at Chicago, Chicago, IL 60612, USA; dgracehu@gmail.com (D.-P.H.); lishixie@uic.edu (L.X.); ranlilu@uic.edu (R.L.); mabern1@uic.edu (M.A.); gprins@uic.edu (G.S.P.); 2Department of Pathology, University of Illinois at Chicago, Chicago, IL 60612, USA; lnonn@uic.edu; 3Massachusetts General Hospital Center for Cancer Research and Harvard Medical School, Charlestown, MA 02129, USA; SHIODA@helix.mgh.harvard.edu

**Keywords:** prostate cancer, stem cells, keratin, RNA-seq, single cell, lineage hierarchy, differentiation

## Abstract

Single prostate stem cells can generate stem and progenitor cells to form prostaspheres in 3D culture. Using a prostasphere-based label retention assay, we recently identified keratin 13 (*KRT13*)-enriched prostate stem cells at single-cell resolution, distinguishing them from daughter progenitors. Herein, we characterized the epithelial cell lineage hierarchy in prostaspheres using single-cell RNA-seq analysis. Keratin profiling revealed three clusters of label-retaining prostate stem cells; *cluster I* represents quiescent stem cells (*PSCA*, *CD36*, *SPINK1*, and *KRT13/23/80/78/4* enriched), while *clusters II* and *III* represent active stem and bipotent progenitor cells (*KRT16/17/6* enriched). Gene set enrichment analysis revealed enrichment of stem and cancer-related pathways in *cluster I*. In non-label-retaining daughter progenitor cells, three clusters were identified; *cluster IV* represents basal progenitors (*KRT5/14/6/16* enriched), while *clusters V* and *VI* represent early and late-stage luminal progenitors, respectively (*KRT8/18/10* enriched). Furthermore, MetaCore analysis showed enrichment of the “cytoskeleton remodeling–keratin filaments” pathway in cancer stem-like cells from human prostate cancer specimens. Along with common keratins (*KRT13/23/80/78/4*) in normal stem cells, unique keratins (*KRT10/19/6C/16*) were enriched in cancer stem-like cells. Clarification of these keratin profiles in human prostate stem cell lineage hierarchy and cancer stem-like cells can facilitate the identification and therapeutic targeting of prostate cancer stem-like cells.

## 1. Introduction

Resident stem cells are a rare cell population found in adult tissues that act as a repair system by maintaining normal regenerative turnover [1,2]. Prostate epithelial stem cells have been identified in the human prostate gland and are thought to generate differentiated epithelial cells that maintain glandular homeostasis throughout life [3,4]. Three types of prostate epithelial cells comprise the simple columnar epithelium: apical secretory luminal cells, underlying basal cells adjacent to the basement membrane, and a small number of neuroendocrine cells [5]. While the differentiation of prostate stem cells into basal, luminal, and neuroendocrine cells has been documented in both the rodent and human prostate [3,4,6,7,8,9,10], the lineage hierarchy for prostate stem cell self-renewal, progenitor cell proliferation, and differentiation remains poorly understood.

Several prostate epithelial cell lineage differentiation models have been proposed. Phenotypic intermediate-type cells, aka the transit-amplifying cells that co-express basal and luminal markers, have been observed in vitro and in vivo [8,9,10]. Therefore, the traditional linear hierarchy model proposes that self-renewing prostate stem cells undergo asymmetric cell division, giving rise to common daughter progenitor cells with high proliferative potential [11,12,13]. In response to signals from the stem cell niche, these cells enter early differentiation pathways to eventually form separate basal, luminal, and neuroendocrine cells [14,15,16]. This suggests that basal and luminal cells are hierarchically related through common progenitor cells. On the other hand, the bifurcated model suggests that basal and luminal progenitor cells represent separate epithelial lineages originating from a common stem cell [17,18].

In a 3D culture system, a single prostate stem cell can undergo both symmetric and asymmetric cell division by self-renewal, generating relatively quiescent daughter stem cells and rapidly proliferating daughter progenitor cells to form spheroids, also known as prostaspheres [19,20,21,22]. Thus, prostaspheres in 3D culture provide a useful system to interrogate the stem cell lineage hierarchy. Recently, our laboratory established a prostasphere-based label retention assay with the capability to identify and isolate long-term label-retaining prostate stem-like cells from their non-label-retaining daughter progenitor cells [23,24]. This approach further enhanced the spheroid model for tracking stem and daughter progenitor cell populations and permitted the functional and transcriptomic analysis of these distinct populations using primary epithelial cells from disease-free human prostates. Of note, the highest differentially expressed gene for the stem cell population was keratin 13 (*KRT13*) [23,24]. Keratins are a family of intermediate filament proteins that perform both mechanical and non-mechanical functions [25,26,27]. Several distinct keratins have been identified in stem cells and cancers where they may play unique roles [28,29,30,31]. Previous studies had identified *KRT13* as a marker for leading edge prostate cancer regional invasion in organ-confined disease [32,33], and our findings confirmed it as a specific human prostate stem cell marker regulating its self-renewal [23,24].

In the present study, we employed the label-retention spheroid assay combined with the Fluidigm C1 and 10X Genomics platforms for single-cell isolation and RNA-seq to identify distinct cell types in the heterogeneous label-retaining stem-like and non-label-retaining progenitor cell populations. Single cell gene profiling combined with bioinformatic tools permitted the clarification of the normal prostate epithelial stem and progenitor cell lineage hierarchy from disease-free human prostate specimens. Specifically, RNA-seq analysis revealed a select group of keratins enriched in label-retaining prostate stem cells, while a separate and distinct group of keratins were enriched in non-retaining basal and luminal progenitors. Clustering analysis, keratin profiling, combined with pseudo-temporal reconstruction and ordering of single cells [34,35], revealed the lineage hierarchy of normal prostate epithelial cells: quiescent prostate stem cells (*cluster I*) become active stem cells (*cluster II*) to generate bipotent progenitor cells (*cluster III*), which give rise to unipotent basal (*cluster IV*) progenitors as well as early (*cluster V*) and late (*cluster VI*) stage luminal progenitors. Using a similar approach with a prostate cancer specimen from organ-confined disease, we identified gene profiles for prostate cancer stem-like cells (cSCs) [23,24] with keratins common to benign stem cells as well as cancer stem-like cell unique keratins. Clarification of the stem cell hierarchy and keratin profiling of human prostate stem cells and cSCs has significant potential for translational research as it permits the identification of novel biomarkers and opens opportunities for developing novel therapeutic strategies that target the cSC population to limit tumor growth and progression.

## 2. Results

### 2.1. Differentially Expressed Keratin Genes in Prostate Stem-Like and Progenitor Cells by Fluidigm Single Cell RNA-Seq

Both structural and non-structural functions of keratins have been identified, and several keratins are associated with stem cells and cancers [25,26,27,28,29,30,31]. Furthermore, keratin profiling has been previously used to study prostate development and epithelial cell differentiation [28,29,30]. To investigate the keratin gene profiles in prostate stem and progenitor cells, we pre-labeled pooled primary prostate epithelial cells (PrECs) from three normal prostate tissues with BrdU or 5(6)-carboxyfluorescein N-hydroxysuccinimidyl ester (CFSE) followed by wash-out in 3D Matrigel culture for 5 days during spheroid formation. Prostaspheres were dispersed into single cells, and CFSE^High^ prostate stem-like cells (1.01%) were separated from CFSE^Low^ progenitor cells (97.52%) by FACS (Figure 1A) [23,24]. The cells were captured using the Fluidigm C1 platform that although isolating and tagging fewer single cells than other systems, permits greater transcriptome resolution with superior coverage across transcripts including gene isoforms. Initial heatmap clustering of 37 keratin genes in the two sorted populations revealed enrichment of *KRT13*, *23*, *80*, *78*, *86*, and *4* in the CFSE^High^ prostate stem-like cells, while *KRT6*, *17*, *14*, *5*, *8*, and *18* were enriched in CFSE^Low^ progenitor cells (Figure 1B). Immunostaining confirmed the enrichment of keratins 13, 23, 4, 78, 80, and 19 at the protein level in BrdU label retaining or CFSE^High^ prostate stem cells (Figure 1C). To better identify the cell subpopulations, we undertook heatmap clustering with selected known gene markers for stem, basal, luminal, and neuroendocrine cells (Figure 1D). Enrichment of the stem cell marker gene *KRT13* was confirmed in CFSE^High^ label-retaining prostate stem cells, while basal marker genes *KRT5*, *KRT14*, *KRT19*, *TP63*, *SOX9*, *GSTP1*, luminal genes *KRT8/18*, and neuroendocrine gene *SYP* were enriched in CFSE^Low^ non-label-retaining prostate progenitor cells, suggesting a stem cell-driven epithelial cell lineage commitment during prostasphere formation. Of note, transcription factor *SOX9* was enriched in both label-retaining stem cells and non-retaining progenitor cells.

### 2.2. Identification of Three Subpopulations in Prostasphere CFSE^High^ Prostate Stem-Like Cells by Single-Cell Analysis

In response to extrinsic signals from the niches, stem cells residing in the quiescent state become activated and enter the cell cycle to progress to lineage-committed daughter progenitor cells [36,37,38]. As such, we hypothesized that CFSE^High^ cells may represent a heterogeneous mix of stem cells at different quiescent and activated stages. Indeed, Fluidigm single-cell RNA-seq of 172 CFSE^High^ cells followed by uniform manifold approximation and projection (UMAP) and Tool of Single Cell Analysis (TSCAN) identified three major clusters (Figure 2A,B), which were further confirmed by unsupervised heatmap clustering (Figure 2C). Cell hierarchical clustering (*cluster I → II → III*) was reconstructed by a minimum spanning tree (MST) incorporated with the traveling salesman problem algorithm to minimize the distance linking cell clusters. A pseudo-temporal ordering score was used to determine starting and ending points of the tree (Figure 2B), which closely mimics the true biological time [35]. Based on these analyses, *cluster I*, the originating cells, are predicted to be quiescent stem-like cells, yielding *cluster II*, likely active stem cells that lead to *cluster III*, which are proposed as bipotent progenitor cells.

Genes previously associated with prostate stemness, e.g., *PSCA* [39], *SPINK1* [40], as well as stemness genes not previously reported in prostate stem cells, e.g., *CD36* [41] were enriched in CFSE^High^ *cluster I* cells. Temporal tracing of these genes by TSCAN positioned them in *cluster I* with decreasing expression through *clusters II* and *III* (Figure 2D), confirming the stem cell nature of *cluster I* cells and the power of this approach.

### 2.3. Keratin Gene Profiling Distinguishes the Prostate Stem and Bipotent Progenitor Cell Clusters

Heatmap clustering of 21 keratin genes in 172 CFSE^High^ sorted prostasphere cells was next used to further distinguish the three cell clusters found in the label-retaining stem-like cell population. Keratin gene profiles revealed the enrichment of *KRT13*, *23*, *80*, *78*, and *4* in *cluster I* stem cells (Figure 3A, middle). In contrast, these keratins were sharply decreased in both *clusters II* (left) and *III* (right), permitting easy identification of the proposed quiescent stem cells*. Clusters II* and *III*, proposed as active stem cells and bipotent progenitors, respectively, were further differentiated by increased *KRT16(16P2*, *16P4*, *16P5)*, *17(17P2*, *17P3*, *17P6)*, and *6* gene expressions in *cluster III*, which were not heightened in *cluster II* (Figure 3A). TSCAN was next employed to trace relative keratin gene expression levels in the three clusters in pseudo-time from *cluster I → II → III*, which is graphically illustrated by circle size for each denoted cell (Figure 3B). Enrichment of *KRT13*, *80*, *78,* and *23* was confirmed for *cluster I* quiescent stem cells, while *KRT16P4 and 17P6* were enriched in *clusters III* bipotent progenitor cells. Active stem cells in *cluster II* express decreased levels of stemness keratins *KRT13*, *80*, *78,* and *23* enriched in *cluster I* with increased levels of *KRT16 (16P2*, *16P4*, *16P5)* and *17 (17P2*, *17P3*, *17P6)* enriched in *cluster III*. Together, these sequential changes in keratin genes as the stem cells enter lineage commitment further supports the progression of cell hierarchy from *cluster I → II*
*→ III.* Furthermore, gene set enrichment analysis (GSEA) revealed the enrichment of stem-cell-related Notch, Toll-like receptor, autophagy, and lysosome pathways in *cluster I* vs. *cluster III* (Figure 3C). These are pathways previously associated with stemness in the prostate and other tissues [42,43,44]. Collectively, these data demonstrate that the keratin profiles rapidly change as the stem cells lineage commit from quiescence to active stem cells and bipotent progenitor cells.

### 2.4. Heatmap Clustering Analysis Identifies Three Separate Clusters of Prostasphere Progenitor Cells with Differentially Expressed Keratin Genes

In the human prostate, epithelial stem cells replenish the tissue and maintain glandular homeostasis by generating lineage-committed progenitor cells that generate basal, luminal, and neuroendocrine cells [10,11,45]. As it is likely that the stem cell-derived, daughter bipotent progenitor cells in prostaspheres initiate and progress along this lineage commitment, we sought to classify the separate progenitor cell populations in the non-label retaining cell population. We FACS separated CFSE^Low^ cells, captured 197 individual cells, and sequenced their transcripts using the Fluidigm scRNA-seq platform. Heatmap clustering analysis for the keratin genes revealed three distinct subpopulations of progenitor cells in the CFSE^Low^ cell fraction (Figure 4A). *Cluster IV* cells (on right) were highly enriched in *KRT5*, *6*, *14*, and 1*6* and are predicted to be unipotent basal progenitor cells. In contrast, *clusters V* and *VI* (middle and left, Figure 4A) were highly enriched in *KRT8*, *18*, and *10* with negligible *KRT5*, indicating a luminal progenitor cell lineage. As *cluster V* retains *KRT16*, a marker for the label-retaining bipotent progenitor (*cluster III*, Figure 3A), and since *KRT8* and *18* are slightly lower in *cluster V* compared to *cluster VI* shown by TSCAN (Figure 4B), *cluster V* is likely an early-stage luminal progenitor, while *cluster VI* represents luminal progenitors further along the lineage commitment pathway. This is further supported by the lowest expressions of *KRT14, 6A,* and 6*C* in *cluster VI* as compared to *clusters IV* and *V*. TSCAN analysis for individual keratin genes in CFSE^Low^ cells were non-linear, representing two separated cell lineages (basal and luminal) at this stage, thus providing no consistent evidence for the derivation of luminal progenitors from basal progenitors (Figure 4B).

### 2.5. RNA-Seq Reveals Differentially Expressed Genes in Cancer Stem-Like Cells

Cancer stem cells often share similar gene marker signatures/signaling pathways with embryonic stem cells and normal adult stem cells [46,47]. To identify common and unique stem cell genes in prostate cancer, primary prostate cancer (PCa) cells were cultured from punch biopsies of three patient specimens collected at prostatectomy with adjacent sections confirmed as 80–100% cancer cells by a board-certified pathologist. These were transferred to 3D culture to generate tumor spheroids using the sphere-based BrdU or CFSE label-retaining system. Cancer stem-like cells (cSCs, CFSE^High^) and non-stem cancer cells (non-cSCs; CFSE^low^) were separated by FACS (Figure 5A), and cDNA libraries were prepared using protocols developed for low cell numbers [48], deep RNA-seq was performed with Illumina NextSeq 500 at the Center for Cancer Research at Harvard Medical School. Unsupervised hierarchical clustering of all transcripts identified 1310 differentially expressed genes (*Q* value < 0.05); 916 enriched in cSCs, and 394 genes enriched in the non-cSC population (Figure 5B). Top enriched genes in cSCs included stem-cell-related genes *PSCA*, *CD36*, and *SPINK1*, among others not previously associated with prostate cancer stem-like cells (Table 1). Heatmap profiles of *CD36*, *KRT13*, *SPINK1*, and *SCGB2A1* document their enrichment in cSCs vs. non-cSCs populations (Figure 5C). Furthermore, immunocytochemistry (ICC) of the PSCA, CD36, and SPINK1 proteins (Figure 5D) confirms their colocalization in the BrdU-labeled cSCs within the tumor spheroids.

Similar to normal prostate stem cells, the cSCs in tumor spheroids exhibited decreased E-cadherin, elevated LC3, an autophagy marker, and *KRT13* (Figure 5D) [23,24,49]. Importantly, the knockdown of *KRT13* by siRNA in primary cells from one patient specimen significantly decreased the tumor spheroid formation, likely by inhibiting cancer stem-like cell self-renewal (Figure 5E).

### 2.6. Prostate Cancer Stem-Like Cells Contain Common and Unique Keratin Genes Compared to Normal Prostate Stem Cells

MetaCore pathway analysis of the RNA-seq data from PCa-derived spheroids derived from the three patient specimens revealed “cytoskeleton remodeling–keratin filaments” as the top enriched pathways in the cSCs (Table 2A). In contrast, cell cycle-related pathways were enriched in the non-cSCs population of these tumor spheroids (Table 2B).

Heatmap clustering of 16 keratin genes in the CFSE^High^ and CFSE^Low^ cells from the tumor spheroids further confirmed the enrichment of specific keratin genes in the cSC fraction (Figure 6). In addition to enrichment for keratins found in normal spheroid stem cells (*KRT13*, *23*, *80*, *78*, and *4*), a group of other keratin genes that includes *KRT10*, *19*, *6*, *75*, *16*, *79*, *3*, and *82* were enriched in the cSCs relative to the non-cSC population of tumor spheroids. Thus, while some similarities to normal prostate stem cells are observed in cancer stem-like cells, several other genes and keratins not typical for normal prostate stem cells are expressed, indicating unique gene profiles in this cancer cell population. We propose that these unique cancer stemness keratin profiles likely reflect the cell-of-origin of the initial tumors.

It is also noteworthy that the cSC population was enriched for genes involved in ligand-independent androgen receptor signaling, IGF-1 receptor signaling, autophagy, and HIF-1 targets (Table 2A), thus identifying potential therapeutic targets for this unique cancer cell type.

### 2.7. Single-Cell RNA-Seq Identifies Cancer Stem-Like Cells with their Differentially Expressed Keratin Genes

To further verify the keratin profile in cSC at the single cell levels, day-7 tumor spheroids were established from a PCa patient specimen and dispersed for single-cell capture using the 10XGenomics platform. Cancer cell subpopulations within the spheroid were interrogated by single-cell RNA-seq analysis using Nova-seq. UMAP clustering defined five cell clusters (*cluster 0-4*) of cancer cells in the tumor spheroids (Figure 7A), with *cluster 2* distinctly separated from the other cell types and enriched in stem cell markers including newly identified stemness keratin gene *KRT13* (Figure 7B, purple dots). Similar to bulk RNA-seq results, single cells RNA-seq followed by keratin gene profiling using heatmap (Figure 7C) and dot-plot (Figure 7D) analyses identified a group of keratin genes (*KRT4*, *13*, *80*, *78*, *23*, *10*, *19*, *16*, *6A*, *6B*, *6C*, *15*, and *17*) enriched in the cSC cluster. Among them, *KRT4*, *13*, *80*, *78*, and *23* are common stemness keratins in normal stem cells, while *KRT10*, *18*, *8*, *19*, *6C*, and *16* are unique cSC associated keratins, mostly identified by bulk RNA-seq (Figure 6).

## 3. Discussion

Overall, the present results show that prostate stem cells can undergo both symmetric and asymmetric cell division in a 3D culture system giving rise to daughter stem and progenitor cells that lineage commit to separate basal and luminal progenitors, primed for further differentiation. Prostate epithelial stem cells play critical roles during prostate development and glandular tissue replenishment. Prostate stem cells generate progenitor cells that transiently amplify and differentiate to mature epithelial lineages [11,12,13]. During prostate epithelial cell differentiation, evidence suggests that there may be a common precursor stem cell for all lineages [14,15,16], while other findings support distinct basal and luminal stem cell populations within the adult prostate [17,18]. In a murine model, Shen et al. reported that single luminal epithelial progenitors can generate prostate organoids in culture [50]. Gao et al. have shown that murine prostate basal cells undergo both symmetric and asymmetric divisions, leading to different cell fates that contribute to the luminal population and tumorigenesis, while luminal cells only exhibit symmetric divisions [12]. Yet these scenarios are not mutually exclusive, as emerging data indicate the inherent plasticity and stage/context-specific utilization of stem and progenitor cell populations [11,12,51]. We and others have previously demonstrated the clonal origin of single human prostate stem cell-derived prostaspheres [19,20]. More recently, using a prostasphere-based label retaining assay, we identified rare prostate stem cells in human specimen-derived prostaspheres and determined that the vast majority of cells are daughter progenitor cells or their progeny [23,24]. These stem cell models provide an excellent opportunity to trace the lineage of prostate epithelial cells differentiation in human samples. Furthermore, single cell RNA-seq analysis now permits the identification of the stem and progenitor subpopulations [34,35], while keratin profiling helps to delineate the hierarchical biological time of progenitor cell lineage commitment [29,30]. The results of the present study show that there are distinct subpopulations of both stem and progenitor cells at different stages in the epithelial lineage hierarchy of the normal prostate and indicate that committed basal and luminal progenitor cells are derived from a common prostate stem cell.

Keratins are useful biological markers for determining the origin and differentiation status of specific epithelial cells in normal tissues as well as their malignant counterparts [28,29,30,52,53]. Keratins serve both mechanical and non-mechanical cellular functions and have been recognized as regulators of various cellular properties and functions, including apicobasal polarization, motility, cell size, protein synthesis, and membrane traffic and signaling [25,26,27]. Using a series of human prostate autoptic tissues of various gestational ages and immunostaining with a comprehensive panel of keratin antibodies, Trompetter et al. identified luminal-type keratins 7, 8, 18, 19, and 20 as well as basal/squamous-type keratins 5, 6, 13, 14, and 17 in the developing fetal human prostate. They proposed that prostate stem cells may contain only keratin 8 but not 5 or 14 [29].

In this study, to monitor stem cell differentiation, we evaluated the expression patterns of keratin genes along with other stem cell markers to classify each subpopulation in the prostate stem and progenitor cell pool. *KRT13*, *23*, *80*, *78*, *86*, and *4* were enriched in label-retaining prostate stem cells, while *KRT6*, *17*, *14*, *5*, *8*, *18*, and *P63* were enriched in non-label retaining progenitors. Importantly, single-cell sequencing permitted the identification of the dynamic process of lineage progression within the label-retaining stem-like cell population, as the expression of these and additional keratins rapidly shifted upon early cell division, allowing the temporal mapping of the prostate stem cell lineage hierarchy from quiescent to active stem cells and the bipotent progenitor cell stages. Similar keratin profile shifts were found in the non-labeled progenitor population, enabling the identification of the distinct basal and luminal cell progenitor populations. It is important to note that cells in day 5 spheroids do not express detectable androgen receptor, HOXB13, or PSA, indicating that the model system represents the early stages of lineage commitment and not full differentiation [19,21,23,24]. Based on these new data, we propose a model of human prostate stem cell hierarchy in the adult gland (Figure 8). Triggered by extrinsic signals from the stem cell niche, quiescent prostate stem cells enriched in *KRT13, 23, 80, 78,* and *4* (*cluster I*) enter the cell cycle and upon division become active stem cells (*cluster II*) with a concomitant decrease in expression of these stemness keratins. With subsequent cell division that triggers increased *KRT 16, 17,* and *6* expression, active stem cells generate bipotent progenitor cells (*cluster III*). The transient amplification of bipotent progenitor cells gives rise to unipotent basal progenitors (*cluster IV*: enriched in *KRT5, 14, 6,* and *16*), as well as early stage luminal progenitors (*cluster V*: enriched in *KRT8, 18,* and *10*, with medium expression of *KRT5, 14, 6*, and *16*) that further lineage progress to late-stage luminal progenitors (*cluster VI*: enriched in *KRT8, 18,* and *10,* with the lowest levels of *KRT5, 14, 6,* and *16*) upon continued division. Thus, common bipotent prostate progenitor cells give rise to committed unipotent basal and luminal progenitor cells, which further supports terminal differentiation toward both lineages (Figure 8). It is important to point out here that these separated cell populations—with stemness activity—have not been previously interrogated in this manner, so this is the first report of putative quiescent and active stem cells as well as bipotent progenitor cells for the human prostate gland. As such, there are no known markers for these three populations to use for confirmatory studies. Enriched genes in these separate populations are currently being interrogated in depth in our lab and will be the subject of a future publication.

Keratins are also used extensively as diagnostic tumor markers, as epithelial malignancies largely maintain the specific keratin patterns associated with their respective cells of origin [28,52,53]. It is widely accepted that most cells in human prostatic adenocarcinomas have keratin phenotypes resembling those of luminal cells but not basal cells, and the lack of basal keratins is a key diagnostic marker for pathologists [30]. LNCaP, an androgen-dependent prostate cancer cell line, expresses *KRT18* but not *KRT5* [30]; however, androgen-independent lines DU145 and PC3 both express basal cell keratins *KRT5*. The expression of *KRT5* in the absence of *KRT14* identifies the existence of an intermediate cell population in prostate carcinomas, and the number of *KRT5*-expressing cells increases significantly after androgen deprivation therapy [54]. Furthermore, aberrant cytoplasmic expression of basal cell marker p63 is associated with increased PCa stem cells as well as PCa mortality [55,56]. As such, it is suggested that the re-expression of basal cell markers including basal keratins may represent the progression to an invasive stage of therapy-resistant PCa. Stem cells are closely associated with tumorigenesis in PCa [57,58,59,60,61], and cSCs are largely enhanced during PCa progression and after conventional therapy [62,63,64]. Thus, increased basal cell markers and keratin expression during PCa metastasis and after androgen deprivation therapy may be an indicator of cSCs expansion. This is supported by our current findings that cSCs express both stemness and basal keratin genes in PCa. We herein demonstrate the enrichment of the “cytoskeleton remodeling–keratin filaments” pathway in cSCs from PCa specimens, identifying both normal stemness genes (*PSCA*, *CD36*, and *SPINK1*) and stem cell keratins (*KRT13*, *23*, *80*, *78*, and *4*) as well as distinctive keratins (*KRT10*, *19*, *6C*, and *16*) in the cSC population. Thus, this profiling allows a reliable gene expression fingerprint to identify the stem-like population in PCa specimens and assess unique and/or enriched genes that can be used for stem-cell isolation, as biomarkers, and for druggable targets to eliminate cancer stem-like cells that are classically resistant to hormone and chemotherapeutic therapeutic modalities.

Taken together, the detailed clarification of the prostate epithelial cell lineage hierarchy by single cell RNA-seq analysis and keratin profiling of human prostate stem cells and cSCs provides abundant opportunities for future translational studies that therapeutically target stem cell populations in both benign prostatic disease and PCa.

## 4. Materials and Methods

### 4.1. Prostate Epithelial Cells and Prostasphere Culture

Primary normal human PrECs were obtained from young (19–21 years of age) disease-free organ donors (Lonza, Walkersville, MD, USA) at passage 2–3 and cultured at 37 °C, 5% CO_2_ in ProstaLife Epithelial Cell Growth Medium (PrEGM) (LifeLine Cell Technology, Carlsbad, CA, USA) on fibronectin-coated flasks. PrECs from four donors were used, and all exhibited similar growth and hormone responsive behaviors. Prostaspheres were cultured from PrECs as previously described and confirmed to be clonally derived spheroids of stem/progenitor cells [19,21,22]. Briefly, 1 × 10^5^ PrEC cells were resuspended in 1:1 Matrigel (Corning/BD Biosciences, Corning, NY, USA)/PrEGM with 1 mL PrEGM and cultured at 37 °C in 5% CO_2_.

Primary human prostate cancer epithelial cells (PrE-Ca) were isolated from radical prostatectomy tissue from a prostate cancer patient at the University of Illinois at Chicago Medical Center, as previously described [38]. Briefly, fresh tissue from the peripheral zone was selected and excised with a 5 mm punch by a pathologist according to an Institutional Review Board-approved protocol. Final tissue pathology was determined by H&E of a thin slice of the punch, and the area was confirmed to be 100% cancer. PrE-Ca cells were maintained in PrEGM medium.

### 4.2. Prostasphere-Based Label Retention Assays

Prostasphere-based label retention assays were performed based on the protocol recently established in our lab [23,24]. For prostasphere-based BrdU retention assays, parental PrECs or primary prostate cancer cells were cultured in 2D in the presence of 1 µM BrdU for 10 days. Then, pre-labeled prostate cells were transferred to 3D Matrigel prostasphere culture for 5 days (≈6 cell cycles with a doubling time of 20 h) in the absence of BrdU to wash-out. The prostaspheres were harvested from Matrigel by dispase digestion on day 5 and plated on chamber slides for overnight culture, allowing spheroid cells to attach and grow outwards on the chamber slides. Then, cells were fixed in ice-cold acetone/methanol (1:1), and BrdU-retaining cells were identified by immunostaining using a mouse anti-BrdU antibody. Images were taken using a Zeiss Axioskop 20 fluorescent microscope, and the numbers of prostasphere clusters and BrdU-positive labeled cells were counted.

For the CFSE label-retention assay (100% overlap with BrdU) in live cells [23,24], 2D-cultured parental PrECs or primary prostate cancer cells were labeled by 5 µM CFSE (Life Technologies, Grand Island, NY, USA) for 30 min. Labeled PrECs were plated in 3D Matrigel in the absence of CFSE and cultured for 5 days to form prostaspheres. Prostaspheres were harvested as above and dispersed into single cells followed by FACS sorting to separate CFSE^High^ and CFSE^Low^ cells.

### 4.3. Gene Knockdown by SiRNA

For siRNA knockdown of the *KRT13* gene, PrE-Ca cancer cells (≈70% confluence) in 2D culture were incubated with 40 nM of siControl (scramble) or si*KRT13* (IDT, Coralville, IA, USA) for 6 h, respectively. Then, 1 × 10^4^/well of siRNA treated PrE-Ca cells with or without CFSE labeling were resuspended in 1:1 Matrigel/PrEGM plated in a 24-well plate for 7 days to form tumor sphere, and the medium was replenished with fresh 40 nM siRNA on day 3 of 3D culture. Tumor sphere numbers and the CFSE^High^ % cells were evaluated as described previously [23,24].

### 4.4. Next-Generation Bulk RNA-Sequencing Analysis

CFSE-labeled PrEC cells from three healthy donors and PCa cells from three patient specimens were used to grow prostaspheres or tumor spheres in 3D cultures. CFSE^High^ and CFSE^Low^ cells were separated by FACS described in Section 4.5. Transcriptome profiles of CFSE^High^ and CFSE^Low^ cells were determined using next-generation RNA-seq performed by Dr. Shioda at the Center for Cancer Research at Harvard Medical School [48]. Briefly, following total RNAs isolation from CFSE^High^ and CFSE^Low^ cells, cDNA synthesis, and fragmentation, indexed deep sequencing libraries for Illumina NextSeq 500 sequencing were generated, and library quantitation and size distributions were determined using Agilent TapeStation. Paired-end sequencing (75 nt + 75 nt) yielded > 50 million raw reads, which were aligned to the human GRCh38/hg38 reference genome sequence. See Supplemental Methods for further details and bioinformatics analysis.

### 4.5. Flow Cytometry

Analysis of trypsin-dispersed CFSE-labeled prostasphere cells was performed by single-channel FACS (CyAn^TM^ ADP Analyzer, Beckman Coulter Inc., Brea, CA, USA) using the Summit Software polygon tool. Prostasphere cells freshly labeled with 5 µM CFSE before FACS sorting were used as positive control, while unlabeled prostasphere cells were used as negative control. Subpopulations of fractionated CFSE^High^ and CFSE^Low^ cells were gated based on negative and positive controls using the FACS-DiVa software polygon tool and collected by CellSorter (MoFlo XDP, Beckman Coulter) [23,24].

### 4.6. Single-Cell Capture Using the Fluidigm C1-IFC System and Single-Cell RNA-Seq by NextSeq500 Deep Sequencing

PrECs were grown as prostaspheres to enrich and separate the primary stem and daughter progenitor populations. The prostasphere-based CFSE retaining assay [23,24] was used to fluorescently label the relatively quiescent prostate stem cells. CFSE^High^ stem cells and CFSE^Low^ progenitor cell fractions were separated by FACS, and the individual populations were separately captured as single cells using the 96-well C1-IFC microfluidic system (Fluidigm, South San Francisco, CA, USA) at the University of Illinois at Chicago (UIC) Research Resource Center. Captured single cells were lysed and reverse transcribed into bar-coded cDNA in each cell of IFC microfluidic chambers. cDNA was ligated to adaptor sequences to construct deep sequencing libraries. The 5′-end and 3′-end of each single-stranded RNA transcript were ligated to different bar-coded adaptors to conserve strand information. Pooled, bar-coded, single-cell cDNA samples were transferred for RNA-seq using the Illumina NextSeq500 deep sequencer at the UIC Genomics Core. Each sequencer run generated 400 million reads, 0.5–2 million reads (paired-end, 75 nt + 75 nt) for each single-cell mRNA sample. We validated ≈150 single-cell cDNA libraries from each sample and ≈10,000 genes per cell, which covers enough coding mRNA for data analysis.

### 4.7. Single-Cell Capture by the 10XGenomics Platform and Single-Cell RNA-Seq Using NovaSeq 6000

Tumor spheroids derived from primary human PCa epithelial cells were cultured for 7 days and dispersed into a single-cell suspension. Single cells were separated and captured at a 10X Genomics station using Chromium Next GEM Single Cell 3ʹ v3 Reagent Kits (10XGenomics, CG000183). After GEM generation and barcoding, mRNA was reverse transcribed into cDNA, which was further amplified for library construction followed by RNA-seq using NovaSeq 6000 SP at the University of Illinois Keck Center. Each lane of the sequencer generated 1.6 billion reads, 0.4 billion reads per sample, 0.1 million reads (paired-end, 28 nt + 91 nt) for each single-cell. We validated ≈3500 single-cell cDNA libraries from each sample and ≈7500 genes per cell for data analysis.

### 4.8. Single-Cell Data Analysis

For the Fluidigm C1 system, RNA-seq data generated from single cells were analyzed by bioinformatics specialists at the UIC Center of Research Informatics. Generated nucleotide base sequences were aligned on the human hg38 reference genome sequence using TopHat2. QC-passed data underwent annotation analysis by cufflinks to calculate FPKM (fragments per kilobase exon/million mapped fragments), reflecting expressed transcript amounts. The normalized FPKM of all single cells was analyzed using the Monocle algorithm [34], which is an unsupervised method with increased temporal resolution of transcriptome dynamics designed for single-cell RNA-seq data analysis. TSCAN was used to analyze temporal transcript dynamics of a pseudotime stem→progenitor cell process to detect genes contributing to cell identity [35]. Hierarchical clustering was used to differentiate meaningful cell signatures. Monocle analyzes the temporal transcript dynamics of the pseudo-time prostate stem → progenitor cells process to detect genes contributing to cell identity. Heatmaps and dendrograms of hierarchical clustering of gene signatures distinguish diverse cell types and identify cell type-specific genes. Enrichment of differentially expressed genes was examined using GSEA. MetaCore Pathway Analysis was used to identify the top enriched pathways across cSC populations.

GSEA was performed on normalized gene expression data from CFSE^High^ and CFSE^Low^ samples generated by RNA-seq to extract biological knowledge using the Broad Institute’s Molecular Signatures Database [65]. Analyses were run using the gene expression data against two gene sets: C2: curated gene sets and CP: BIOCARTA: BioCarta gene sets. The false discovery rate (FDR) was calculated by comparing the actual data with 1000 Monte Carlo simulations. The familywise error rate is a conservative correction that seeks to ensure that reported results do not include any false-positive gene sets.

For the 10XGenomic platform, single-cell RNA-seq data generated by NovaSeq 6000 was aligned against hg38 reference genome sequence using 10x Cellranger pipeline. The aligned dataset was imported into Seurat package (Satijalab). Pre-processing of raw data included filtering out cell debris, unhealthy singlets, and potential doublets by setting cutoff based on the distribution of the feature RNA abundance (cutoff range was from 2000 to 7500; each data point indicates a singlet or single cell) and the percentage cutoff of mitochondrial genes within each data point was 10%. Post-filtered single cell feature expression values were normalized by the total expression through a global-scaling normalization method “LogNormalize”. The scale factor was set at the default value of 10,000. Subsequent feature selection was performed with default setting (nfeature = 2000) to gain a subset of features that reflect high cell-to-cell variation of the dataset. The expression value of each gene was scaled so that data points were centralized around the origin, and the variance across these data points equals to 1. Linear dimensional reduction was performed on the subset of feature genes selected by PCA. Next, a resampling test inspired by the Jackstraw procedure was utilized to determine the PC number to include. The data points or single cells were finally clustered based on PCs determined previously using a graph-based clustering approach based on Macosko et al. [66]. Pre-defined PCs were introduced to the clustering analysis to plot UMAP for the better visualization of clusters. Subsequent identification of each cluster was performed mainly by keratin profiling and according to canonical biomarkers [67].

### 4.9. Immunocytochemistry (ICC)

ICC was performed on prostaspheres attached overnight on chamber slides (Millipore, Billerica, MA, USA) to permit limited cell outgrowth and improve reagent penetration [21,22,23,24]. Spheres were fixed in ice-cold acetone/methanol (1:1) and blocked in 5% normal goat serum in PBST (PBS with 0.25% Triton X-100) for 30 min at room temperature. After acid wash with 2N HCl for 30 min at room temperature, spheres were incubated with primary antibodies diluted in 2% normal goat sera in PBST overnight at 4 °C, rinsed 3X for 5 min in PBS, and incubated with secondary goat antibody (diluted 1:1000 in 2% normal goat serum in PBST) at room temperature for 2 h. Primary anti-human antibodies used for double staining are anti-BrdU antibody (1:200, #ab152095, Abcam, San Francisco, CA, USA), anti-*KRT13* antibody (1:200, #ab97327, Abcam, San Francisco, CA, USA), anti-*KRT14* antibody (1:200, #prb-155p, Covance Inc., Princeton, NJ, USA), anti-*KRT80* antibody (1:2000, #16835-1-ap, Proteintech, Rosemont, IL, USA), anti-*KRT4* antibody (1:400, Abcam, #ab51599, San Francisco, CA, USA), anti-*KRT78* antibody (1:2000, #PA5-58902, Invitrogen, Waltham, WA, USA), anti-*KRT23* antibody (1:400, #sc365892, Santa Cruz Biotechnology, Dallas, TX, USA), anti-*KRT19* antibody (1:400, #NBP2-22116SS, Novus, Centennial, CO, USA), anti-E-cadherin antibody (1:100, #sc8426, Santa Cruz Biotechnology, Dallas, TX, USA), anti-LC3 antibody (1:200, #12741p, Cell Signaling Technology, Danver, MA, USA), anti-CD36 antibody (1:150, #mab19553-sp, Novus, Centennial, CO, USA), anti-SPINK1 antibody (1:200, #mab7496, R&D Systems, Irvine, CA, USA), and anti-PSCA antibody (1:50, #abx236848, Abbexa, Cambridge, UK). Secondary goat anti-rabbit Alexa Fluor 568 (1:2000, #ab175471, Abcam, San Francisco, CA, USA) was used. Following rinses in PBS, slides were cover slipped with fluorescence mounting medium with DAPI (Vectashield, Vector Laboratories, Burlingame, CA, USA). Mouse and rabbit IgG (eBioscience, San Diego, CA, USA) antibodies were used as negative controls. Images of stained spheres/cells were obtained on a Zeiss Axioskop 20 fluorescent microscope with an Axiocamera or a Zeiss LSM 510 META Confocal Microscope.

### 4.10. Statistical Analysis

Significant differentially expressed transcripts between groups of single cells were identified with Benjamini–Hochberg corrected *p*-value (FDR) for multiple comparisons. Experiments for in vitro studies were biologically replicated 3 times. Data were analyzed using Student’s *t-*test. Values are expressed as mean ± SEM, and *p* < 0.05 was considered significant.

## 5. Conclusions

Our novel sphere-based label retention assay followed by single-cell RNA-seq analysis and keratin profiling provides a unique tool to study prostate epithelial cell lineage hierarchy and cancer stem-like cells. Its broad application further enables discovery of potential biomarkers for prostate cSCs and creates new strategies for managing therapy-resistant prostate cancer.

## Figures and Tables

**Figure 1 ijms-22-08109-f001:**
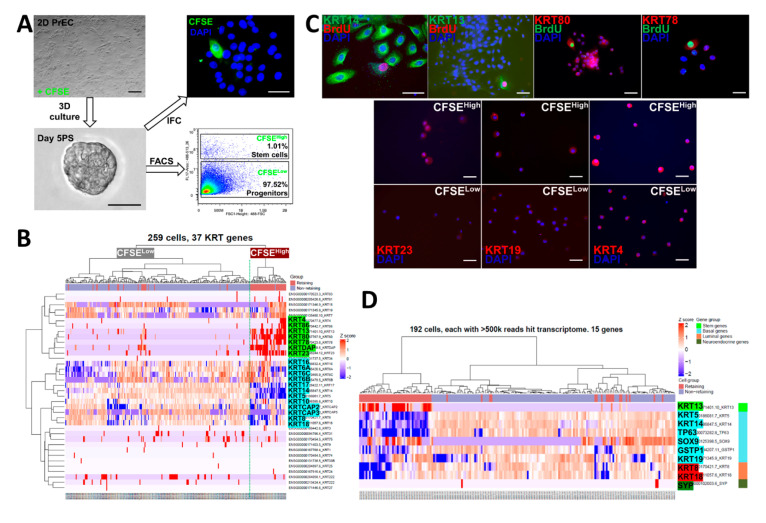
Two groups of differentially expressed keratin genes in CFSE^High^ vs. CFSE^Low^ day 5 prostasphere cells. (**A**) Normal primary prostate epithelial cells (PrEC) pre-labeled with CFSE were 3D cultured (5 days) to form prostaspheres (PS), followed by FACS to separate CFSE^High^ stem-like cells (1.01%) from CFSE^Low^ progenitor cells (97.52%). Scale bar = 50 µm. (**B**) Heatmap clustering of Fluidigm C1 captured single-cell RNA-seq of prostasphere cells revealed enrichment of *KRT13, 23, 80, 78, 86,* and *4* in CFSE^High^ prostate stem-like cells (89 cells), while *KRT6, 17, 14, 5, 8,* and *18* were enriched in CFSE^Low^ progenitor cells (170 cells). (**C**) Top, Immunostaining confirmed the colocalization of keratins 13, 80, and 78 in BrdU label retaining prostasphere cells (pink) and keratin 14 localization in non-labeled progenitor cells. Bottom, FACS isolated CFSE^High^ prostate stem-like cells immunostained for keratin 23, 19, while CFSE^Low^ progenitor cells did not. Scale bar = 50 µm. (**D**) Single-cell RNA-seq of CFSE labeled prostaspheres followed by heatmap clustering using selected genes for stem, basal, luminal, and neuroendocrine cells showed enrichment of stem cell marker gene *KRT13* in CFSE^High^ prostate stem cells, while basal marker genes *KRT5*, *KRT14*, *TP63*, *GSTP1*, and *KRT19*, luminal genes *KRT8/18*, and neuroendocrine gene *SYP* were enriched in CFSE^Low^ prostate progenitor cells. SOX9 was uniquely enriched in CFSE^High^ cells and a subpopulation of CFSE^Low^ cells.

**Figure 2 ijms-22-08109-f002:**
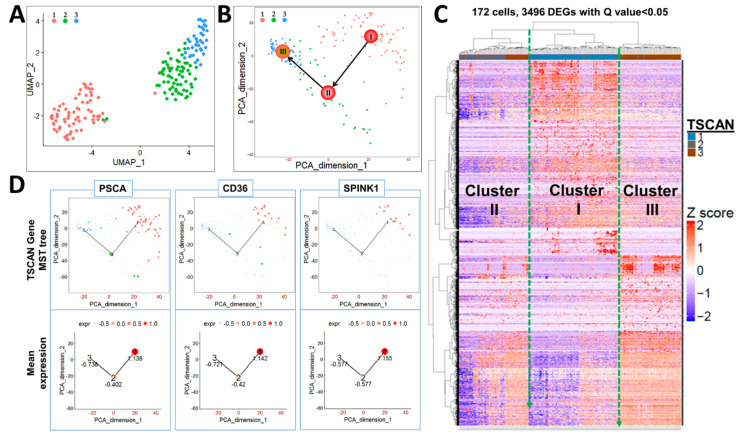
Single-cell RNA-seq of CFSE^High^ cells revealed three subpopulations of cells. Data from Fluidigm C1 single-cell RNA-seq of CFSE^High^ isolated prostasphere cells was analyzed by UMAP (**A**) and TSCAN (**B**) principal component analyses, which identified three major cell clusters that were further confirmed by unsupervised heatmap clustering (**C**). CFSE^High^ cell hierarchical clustering (*I* (1) → *II* (2) → *III* (3)) was ordered by MST-based pseudo-time reconstruction to mimic true biological time, identifying cells in *cluster I* as originating cells which yield *cluster II* that then yield *cluster III* (labeled in B). (**D**) Known stemness genes *PSCA*, *CD36*, and *SPINK1* are enriched in *cluster I* (1) as compared to *clusters II* (2) & *III* (3), which is demarcated by a circle size (top) that represents gene expression levels that are relatively quantified at the bottom.

**Figure 3 ijms-22-08109-f003:**
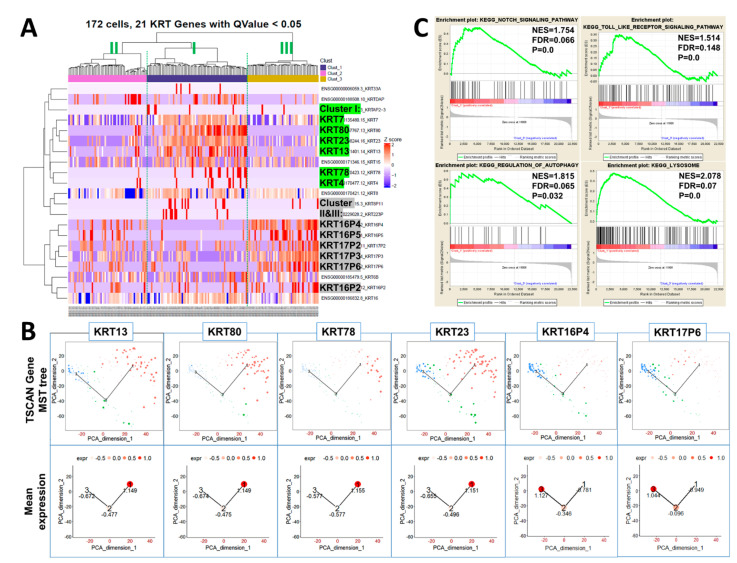
Differentially expressed keratin genes in CFSE^High^ prostasphere cells are sufficient to distinguish cells in *cluster I*, *cluster II*, and *cluster III*. (**A**) In CFSE^High^ cells, *KRT13, 23, 80, 78,* and *4* are enriched in *cluster I* quiescent stem cells and have low expression in *clusters II* and *III*. *KRT16, 17*, and *6* genes are enriched in *cluster III*, whereas *cluster II* has low expression levels of this keratin gene set. (**B**) TSCAN analysis documents enrichment of *KRT13, 80, 78,* and *23* in *cluster I* (1) quiescent stem cells and of *KRT16P4* and *17P6* in *cluster III* (3) cells demarcated by the circle size (top) which is directionally relatively quantified at bottom. (**C**) GSEA analysis identifies enrichment of stem cell-related Notch, Toll-like receptor, autophagy, and lysosome pathways in *cluster I* (1) vs. *cluster III* (3).

**Figure 4 ijms-22-08109-f004:**
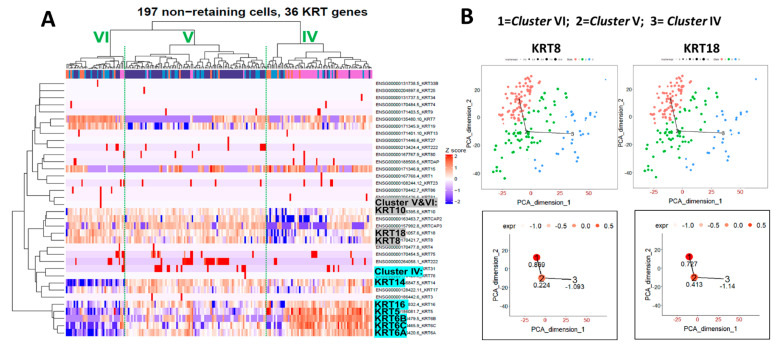
Keratin profiles of differentially expressed keratin genes in non-label retaining prostasphere cells. (**A**) Single-cell RNA-seq of the sorted CFSE^Low^ prostasphere cells with subsequent keratin gene heatmap clustering reveals three subpopulations of progenitor cells: *cluster IV* (right), *cluster V* (middle), and *cluster VI* (left)*. Cluster IV* is enriched in *KRT14, 5, 16,* and *6*, indicating a basal progenitor cell lineage. *Clusters V* and *VI* are enriched in *KRT8, 18,* and *10*, indicating a luminal cell lineage. (**B**) TSCAN analysis of *KRT8* and *KRT18* in CFSE^Low^ cells suggests a relationship between *cluster V* (2) and *VI* (1), but not *cluster IV* (3). Relative gene expression levels are noted numerically for each cluster.

**Figure 5 ijms-22-08109-f005:**
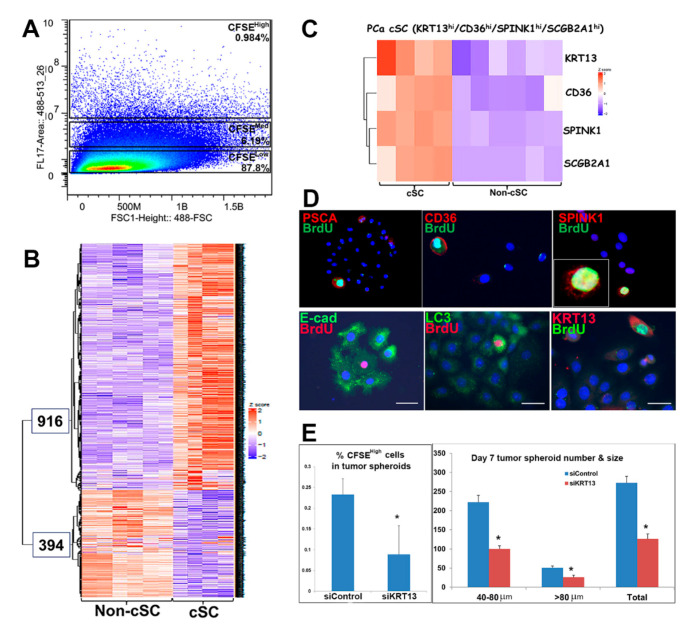
RNA-seq reveals differentially expressed genes in cancer stem-like cells and vs. non-stem cells in tumor spheroids of patient PCa specimens. (**A**) Cancer stem-like cells (cSC; (CFSE^High^) and non-cSC (CFSE^low^) from spheroids of three PCa specimens were separated by FACS. (**B**) Following RNA-seq of the sorted cells, unsupervised transcriptome profiling identified 1310 differentially expressed genes (*Q* value < 0.05); 916 enriched in cSCs and 394 enriched in non-cSCs. Sequencing of the non-cSC fractions from three patients were performed in duplicate, highlighting data consistency. For the cSC fraction, one patient sample was sequenced in duplicate, and the other two were sequenced in singlet. (**C**) Heatmap of *CD36, KRT13, SPINK1*, and *SCGB2A1* expression shows enrichment in cSCs vs. non-cSCs. (**D**) ICC documents that PSCA, CD36, and SPINK1 proteins (red) colocalized with BrdU-labeled (green) cSC cells in the 3D cultured tumor spheroids. Similar to normal prostate stem cells, the cSCs exhibit decreased E-cadherin, increased LC3 and KRT13 (Hu et al., 2017), which together identifies unique features of PCa-derived cancer stem-like cells. (E) siRNA knockdown of *KRT13* significantly decreased the percentage of CFSE^High^ cells in tumor spheroids (left) and tumor spheroid formation (right). * *p* < 0.05 vs. siControl, N = 3.

**Figure 6 ijms-22-08109-f006:**
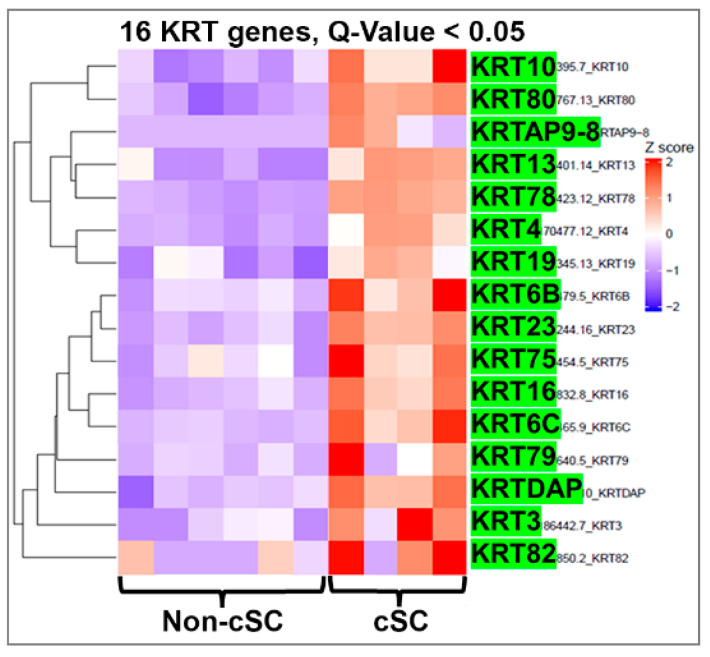
Heatmap clustering of keratin genes in PCa stem-like cells (cSCs) and the non-stem cells in tumor spheroids from PCa specimens. RNA-seq followed by heatmap clustering analysis shows that in addition to normal stem cell-enriched keratins (*KRT13, 23, 80, 78, 4*), *KRT10, 19, 6C, 75, 16, 79, 3,* and *82* also were enriched in cancer stem-like cells relative to expression levels in the non-cSC population. *Q*-value < 0.05.

**Figure 7 ijms-22-08109-f007:**
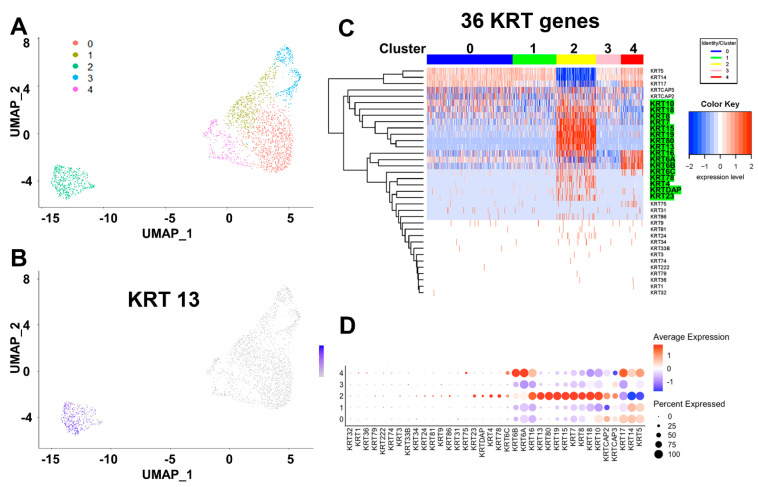
Keratin profiling in tumor spheroids using single-cell RNA-seq identifies cancer stem-like cells. (**A**,**B**) Uniform manifold approximation and projection (UMAP) clustering defined five clusters in cancer cells, identifying *cluster 2* as cancer stem-like cells enriched for stemness markers, including *KRT13*. (**C**) Keratin gene profiling by heatmap and (**D**) dot-plot analyses identified a group of keratin genes enriched in the *cluster 2* cancer stem-like cells.

**Figure 8 ijms-22-08109-f008:**
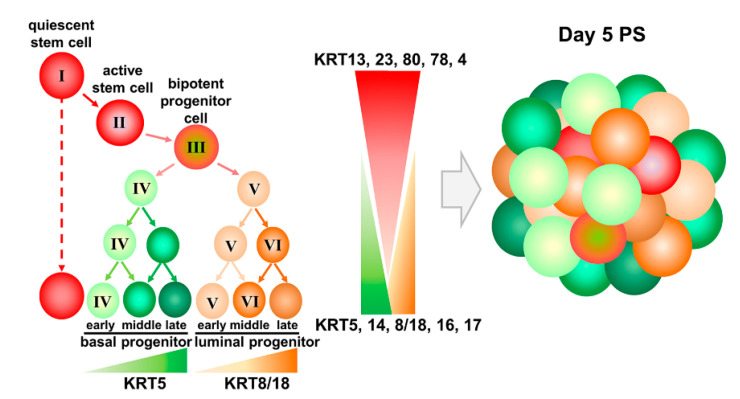
Model of prostate epithelial stem cell hierarchy based upon keratin profiles. Quiescent prostate stem cells (*cluster I*: enriched for *KRT13, 23, 80, 78, 4*) become active stem cells (*cluster II*), with increased expression of *KRT 16, 17,* and *6* that generate bipotent progenitor cells (*cluster III*) and further give rise to committed unique potent basal progenitors (*cluster IV*: enriched for *KRT5, 14, 6, 16*), as well as early (*cluster V*) and late-stage (*cluster VI*) luminal progenitors (enriched for *KRT8, 18, 10*).

**Table 1 ijms-22-08109-t001:** Top genes enriched in prostate cancer stem-like cells.

Gene		cSC/non-cSC
Symbol	Gene Name	(Fold Change)
SCGB2A1	Secretoglobulin family 2A1	67,370
SPINK1	Serine peptidase inhibitor kazak-type 1	2270
PSCA	Prostate stem cell antigen	1621
CD36	Scavenger receptor class B member 3	990
TMPRSS11E	Transmembrane serine protease 11E	593
HMOX1	Heme oxygenase 1	128
IGF2	Insulin growth factor 2	117

**Table 2 ijms-22-08109-t002:** Enriched pathways in cSC cells (**A**) and non-cSC cells (**B**) by MetaCore Enrichment analysis.

(**A**) **Pathways enriched in cSC cells** (Prostate_Cancer_916_genes_*Q*value_0.05_genelist)					
**Maps**	**Total**	***p*-Value**	**FDR**	**In Data**	**Network Objects from Active Data**
**Cytoskeleton remodeling: Keratin filaments**	36	2.7 × 10^−8^	1.8 × 10^−5^	10	Keratin 16, 14-3-3 gamma, Keratin 4, PPL (periplakin), Keratin 6C, Keratin 6A, Keratin 19, Keratin 13, Keratin 4/13, Plakophilin 1
**Ligand-independent activation of androgen receptor in PCa**	67	1.3 × 10^−5^	3.1 × 10^−3^	10	GAB1, Bcl-XL, PP2A regulatory, c-Myc, ERK1 (MAPK3), S5AR2, PP2A catalytic, FGFR1, Kallikrein 3 (PSA), ErbB3
**Androgen receptor activation and downstream signaling in PCa**	110	8.5 × 10^−4^	6.2 × 10^−2^	10	GAB1, Bcl-XL, TMPRSS2, c-Myc, ERK1 (MAPK3), S5AR2, Kallikrein 2, FGFR1, PSCA, Kallikrein 3 (PSA)
**Transcription: HIF-1 targets**	95	5.4 × 10^−5^	0.0071	11	ROR-alpha, TGM2, Mxi1, CITED2, Heme oxygenase 1, HXK2, c-Myc, Cyclin G2, LOXL2, AK3, CTGF
**Development: IGF-1 receptor signaling**	51	0.00044	0.0416	7	Bcl-XL, MNK2(GPRK7), c-Myc, IGF-2, IBP, FOXO3A, ERK1/2
**Autophagy**	35	0.00246	0.0993	5	GATE-16, Bcl-XL, WIPI2, MAP1LC3A, ULK1
(**B**) **Pathways enriched in non-cSC cells** (Prostate_Cancer_394_genes_*Q*value_0.05_genelist)					
**Maps**	**Total**	***p*-Value**	**FDR**	**In Data**	**Network Objects from Active Data**
**Cell cycle: Chromosome condensation in prometaphase**	21	4.4 × 10^−14^	2 × 10^−11^	11	Aurora-B, BRRN1, CAP-H/H2, CAP-G, CAP-E, Cyclin B, CAP-C, TOP2, CAP-G/G2, Histone H1, CDK1 (p34)
**Cell cycle: The metaphase checkpoint**	36	2.3 × 10^−12^	6 × 10^−10^	12	Nek2A, Aurora-B, HEC, CDCA1, CDC20, HZwint-1, Rod, CENP-F, DSN1, SPBC24, CENP-E, PLK1
**Cell cycle: Transition and termination of DNA replication**	27	1.4 × 10^−9^	2 × 10^−7^	9	TOP2 alpha, POLD reg (p68), Brca1/Bard1, Brca1, MCM2, TOP2, FEN1, Bard1, CDK1 (p34)
**Cell cycle: Start of DNA replication in early S phase**	32	2.5 × 10^−6^	0.0001	7	CDC18L (CDC6), MCM10, Geminin, MCM4, MCM2, Histone H1, CDC45L
**Cell cycle: Initiation of mitosis**	26	2.5 × 10^−8^	3 × 10^−6^	8	Lamin B, Cyclin B1, Cyclin B2, FOXM1, Kinase MYT1, Histone H1, CDK1 (p34), PLK1
**Signal transduction: NF-kB activation pathways**	51	5 × 10^−8^	4 × 10^−6^	10	NF-kB2 (p100), IL-1 beta, NF-kB2 (p52), I-kB, NF-kB, c-IAP1, c-IAP2, NF-kB1 (p105), BAFF(TNFSF13B), NF-kB1 (p50)

## Data Availability

Data are available in GEO data portal (GSE95542).

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
