# Peer review of "Keratin Profiling by Single-Cell RNA-Sequencing Identifies Human Prostate Stem Cell Lineage Hierarchy and Cancer Stem-Like Cells"

_ijms, 2021, doi:10.3390/ijms22158109_

Round 1
Reviewer 1 Report
In this manuscript, Wen-Yang Hu and coworkers have employed the prostasphere-based label retention assay to analyze keratin (KRT) profiling and characterize the epithelial cell lineage hierarchy. The work identified different clusters of cells in normal epithelial cells, including quiescent prostate stem cells, active stem cells, bipotent progenitor cells, unipotent basal progenitors, and early and late stages luminal progenitors. The characterization of KRT profiles was performed for all the cell clusters. In addition, the authors showed a similar approach with prostate cancer specimen, which describes interesting new findings of KRT profiles that can reflect the cell-of-origin of the tumors. These new findings can be applied to the identification of novel biomarkers and the development of novel therapeutic strategies. Overall, this is a nice study that advances understanding of prostate stem cell lineage hierarchy and cancer stem-like cells. I only have a few minor questions/suggestions for the authors in revising their manuscript.
- In figure 1-A, the authors performed the cell sorting, separating CFSEHigh prostate stem-like cells (1.01%) from CFSELow progenitor cells (97.52%). Can the authors explain what is threshold for this? As a negative control (no expression), was it used non-treated cells?
- In figure 1-B, it was analyzed 249 cells, but how many cells were analyzed in each subgroup?
- The immunostaining in figure 1-C showed the staining for different KRTs, is it possible to see the staining for KRT14, 13, 80, and 78 after cell sorting? The antibodies for KRT80 and KRT4 seem to stain the CFSELow progenitor cells too. Can the authors explain this?
- In Figure 1-D, SOX9 seems to be enriched in CFSEHigh prostate stem-like cells too. There is an explanation for that?
- In the TSCAN Gene MST tree showed in Figures 2-D and 3-B, is it possible to increase the labels’ size?
- Figure 3-C showed the GSEA analysis, was it showed all the pathways enriched in cluster I vs cluster III. The FDR for the enrichment identified was high. Was analyzed the enrichment in cluster III vs cluster I?
- In Figure 4, is it possible to characterize the expression of the population for KTR16 and 8 by immunostaining?
- On line 209, the authors state “This is further supported by a near total absence of KRT14, and 6A and C in Cluster VI.” The authors should scale down this argument because it seems that there is some expression of KRT6C in Cluster VI.
- In Figure 6, can be performed differential gene expression for CFSEHigh and CFSELow cells from the tumor spheroids vs CFSEHigh and CFSELow cells from the normal spheroids, respectively?
- In Figures 7-A and B, can it be used the same color code for the clusters?
- In section 4.4. - flow cytometry, the authors should provide more details about negative and positive control.
Author Response
We want to thank reviewer for the positive comments and all the great suggestions. We believe that we have addressed all the reviewer comments below, and greatly improved our manuscript before publication. Below are the point-by-point responses.
1. In figure 1-A, the authors performed the cell sorting, separating CFSEHigh prostate stem-like cells (1.01%) from CFSELow progenitor cells (97.52%). Can the authors explain what is threshold for this? As a negative control (no expression), was it used non-treated cells?
This is a great question. The threshold for FACS sorting was determined by setting up the gate using positive (CFSE-labeled) and negative (unlabeled) controls to separate the subpopulation of CFSEHigh cells from CFSELow cells. The thresholds range from 0.5 to 2% depending on the different cell sources and conditions. The typical threshold of 1% reflects the common tissue stem cells subpopulation documented in the majority of stem cell studies. The details were described in our previous publications (Stem Cell Research 2017, 23, 1-12. Fig 1D & 2B; J. Vis. Exp. 2019, 154, e60357. Fig 2,3 &4).
2. In figure 1-B, it was analyzed 259 cells, but how many cells were analyzed in each subgroup?
Thank you. We added in the figure legend that 259 cells included 170 non-retaining CFSELow progenitor cells and 89 retaining CFSEHigh prostate stem-like cells.
3. The immunostaining in figure 1-C showed the staining for different KRTs, is it possible to see the staining for KRT14, 13, 80, and 78 after cell sorting? The antibodies for KRT80 and KRT4 seem to stain the CFSELow progenitor cells too. Can the authors explain this?
Thanks for the comment. There are some advantages in directly using attached spheres for ICC staining without FACS separation. We double stained KRT proteins with BrdU to show the enrichment of KRT proteins in BrdU-label retaining stem-like cells or non-retaining progenitor cells on the same slide, that allow for direct comparison of KRT proteins in both cell populations. We got very nice double staining results for KRT14, 13, 80, and 78, but antibodies for KRT4, 19 and 23 did not work well with BrdU antibody at double staining, that is the reason why we had to separate CFSEHigh cells and CFSELow cells by FACS in order to perform the single antibody staining for these 3 KRT proteins.
4. In Figure 1-D, SOX9 seems to be enriched in CFSEHigh prostate stem-like cells too. There is an explanation for that?
Good question. Transcription factor SOX9 has been shown to play oncogenic role in different types of human cancer, it also acts as a stem cell factor that promotes lineage plasticity and cancer progression. Our single cell RNA-seq data showed the enrichment of SOX9 in both label-retaining stem cells and a subpopulation of non-retaining progenitor cells, as stated in Figure Legend “SOX9 was uniquely enriched in CFSEHigh cells and a subpopulation of CFSELow cells.”
5. In the TSCAN Gene MST tree showed in Figures 2-D and 3-B, is it possible to increase the labels’ size?
Thanks for the suggestion. We increased the labels’ size in the new Figures 2-D and 3-B.
6. Figure 3-C showed the GSEA analysis, was it showed all the pathways enriched in cluster I vs cluster III. The FDR for the enrichment identified was high. Was analyzed the enrichment in cluster III vs cluster I?
We have done the GSEA analysis of the top enriched pathways for cluster III vs cluster I as well. Since we focus on the stem cell signaling in the current study, we only showed the stem cell related pathways enriched in cluster I vs cluster III including Notch, TLR, autophagy and lysosome, which are known stem cell signaling pathways in the prostate based on our previous work (Hu et al, 2017). Different from 0.05 as the classic significant P value cut off, 0.25 is commonly used as an FDR cut off and is a reasonable setting of exploratory discovery where one is interested in finding candidate hypothesis to be further validated as a result of future research. The FDR values of the enriched pathways present in the current study were all well below 0.25.
7. In Figure 4, is it possible to characterize the expression of the population for KTR16 and 8 by immunostaining?
Thanks for the comment. We have validated a list of potential stemness markers by ICC staining using antibodies against selected key keratin proteins identified in the current study. KRT16 and 8 are not the top genes on the list and are less important compared to other top stemness keratin proteins. Also, the editor of the Journal only gave us 10 days to submit the revised manuscript, therefore we are not able to run new ICC staining for KRT16 and 8.
8. On line 209, the authors state “This is further supported by a near total absence of KRT14, and 6A and C in Cluster VI.” The authors should scale down this argument because it seems that there is some expression of KRT6C in Cluster VI.
Thanks for the suggestion. We revised the statement in the text to “This is further supported by the lowest expressions of KRT14, KRT6A and KRT6C in Cluster VI as compared to Clusters IV and V.”
9. In Figure 6, can be performed differential gene expression for CFSEHigh and CFSELow cells from the tumor spheroids vs CFSEHigh and CFSELow cells from the normal spheroids, respectively?
This is a great comment. We have done the differentially expressed genes (DEGs) for CFSEHigh and CFSELow cells from both tumor and normal spheroids; part of this data was published in Stem Cell Research 2017, 23, 1-12 and other data will be included in our future publications. The current study is solely focusing on keratin genes profiling to identify the prostate stem cell lineage hierarchy and cancer stem-like cells. Therefore, we have chosen to only list the differentially expressed keratin genes in cancer stem-like cells versus non-stem cancer cells.
10. In Figures 7-A and B, can it be used the same color code for the clusters?
The 5 different colors in Fig 7A represent 5 different clusters (0-4) of single cells, while the intensity of blue color in Fig 7B represents the different expression levels of KRT13 gene in each single cells. Therefore, we cannot use the same color code for Fig 7-A and B.
11. In section 4.4. - flow cytometry, the authors should provide more details about negative and positive control.
Thanks for the suggestion. We added the detail description in the Method section 4.4 “Prostasphere cells freshly labeled with 5 µM CFSE before FACS sorting were used as positive control while unlabeled prostasphere cells were used as negative control.”
Reviewer 2 Report
Comments
For result 2.1(Figure 1)
- Scale bar in figure 1A is not shown.
- Please show CFSE fluorescence before 3D culture at day 0 and after sorting.
- Although scale bar in figure 1C is shown, the length of scale bar was not mentioned in the figure legend.
- Between CFSEHigh and CFSELow, the brightness of red channel looks different in figure 1C.
- The staining of KRT80, 78, 23, 19 and 4 is not clear in cell membrane in figure 1C.
- In material and method, please describe the product number of antibodies.
- 1(luminal cells) and Chromogranin A (neuroendocrine cells) should be included in figure 1D.
- In the text, please mention about that SOX9 is also enriched in CFSEHigh cells in figure 1D.
- In KRT5, 8 and 18 expressions of CFSELow cells, luminal and basal lineage does not look separated completely figure 1D. Please explain about this.
For result 2.2 (Figure 2)
- The authors defined cluster I as quiescent stem-like cells, cluster II as active stem cells, and cluster III as bipotent progenitor. However, the evidence for defining clusters is not mentioned clearly (line 147-149).
- Markers of active stem cells and bipotent progenitor are also required for cluster definition (Figure 2D).
- In the reference article, SPINK1 is studied in prostate cancer cells. Therefore, if possible, other stemness markers in normal prostate stem cells are desirable.
For result 2.3(Figure 3)
- Please mentions about Cluster II in this part.
For result 2.4(Figure 4)
- The authors should show the analyses with KRT5 and non-keratin markers analysis.
- Time course analysis with prostasphere should be performed to show cell differentiation.
For result 2.5(Figure 5)
- In material and method, protocol of siRNA is not mentioned (related to figure 5E).
- To confirm the inhibitory effect of KRT13 knockdown in stem cell self-renewal, please consider to inhibit KRT13 in CFSEHighcells of prostasphere and culture for 5 days to reconstruct prostaspheres.
For result 2.6(Figure 6)
- In vivo analysis should be performed for confirmation of stem cell identity.
For result 2.7 (Figure 7)
- The authors should show the analyses with KRT5 and non-keratin markers analysis.
- Time course analysis with prostasphere should be performed to track cell differentiation.
For discussion (Figure 8)
- Difference between cluster IV and V was not fully mentioned. Therefore, differentiation from III to IV and V in the model is not clear (Time course analysis of prostasphere reconstruction from CFSEHigh might be useful for confirmation of the model).
- Keratin profiling of prostate cells is interesting for studying development/regeneration of prostate. However, Analysis of keratin profiling with in vitro cultured samples is not sufficient and with in vivo samples is also required.
For some comments
- In line 265, “confirmed the enrichment of of specific”→ “confirmed the enrichment of specific”?
- In line 350, “shifts were foiund in the non-labeled progenitor” → “shifts were found in the non-labeled progenitor”?
For comments in reference
- 23: Hu WY, Hu DP, Xie L, Li Y, Nonn L, Shioda T, Prins GS. →Hu WY, Hu DP, Xie L, Li Y,Majumdar S, Nonn L, Shioda T, Prins GS.
- 59: 67, 4708-4715 → 67, 4807-4815
Author Response
We want to thank reviewer for all the comments and great suggestions. We believe that we have addressed all the reviewer comments and greatly improved our manuscript as a result. Below are the point-by-point responses.
For result 2.1(Figure 1)
Scale bar in figure 1A is not shown.
Thanks for the comment. We added scale bars in Fig 1A and the label “Scale bar=50 µm” in the figure legend.
- Please show CFSE fluorescence before 3D culture at day 0 and after sorting.
The details of CFSE fluorescence imaging have been shown in our previous publications and are cited in this paper as reference #23 and #24 (Stem Cell Research 2017, 23, 1-12. Fig 1D & 2B; J. Vis. Exp. 2019, 154, e60357. Fig 2,3 &4).
- Although scale bar in figure 1C is shown, the length of scale bar was not mentioned in the figure legend.
We added “Scale bar=50 µm.” in figure legend.
- Between CFSEHigh and CFSELow, the brightness of red channel looks different in figure 1C.
The brightness of red channel between CFSEHigh and CFSELow is matched in the new Figure 1C.
- The staining of KRT80, 78, 23, 19 and 4 is not clear in cell membrane in figure 1C.
The keratin proteins are mainly located in cytoplasm, we do not expect the staining of keratin proteins on cell membrane.
- In material and method, please describe the product number of antibodies.
We added the product numbers in the Material and Method section for each antibody used. “Primary anti-human antibodies used for double staining are: anti-BrdU antibody (1:200, #ab152095, Abcam, San Francisco, CA); anti-KRT13 antibody (1:200, #ab97327, Abcam, San Francisco, CA), anti-KRT14 antibody (1:200, #prb-155p, Covance Inc., Princeton, NJ), anti-KRT80 antibody (1:2000, 16835-1-ap, ProteinTech, Rosemont, IL), anti-KRT4 antibody (1:400, Abcam, #ab51599, San Francisco, CA), anti-KRT78 antibody (1:2000, #PA5-58902 , Invitrogen, Waltham, WA), anti-KRT23 antibody (1:400, #sc365892, Santa Cruz Biotechnology, Dallas, TX), anti-KRT19 antibody (1:400, #NBP2-22116SS , Novus, Centennial, CO), anti-E-cadherin antibody (1:100, #sc8426, Santa Cruz Biotechnology, Dallas, TX), anti-LC3 antibody (1:200, #12741p, Cell Signaling Technology, Danver, MA), anti-CD36 antibody (1:150, #mab19553-sp , Novus, Centennial, CO), anti-SPINK1 antibody (1:200, #mab7496, R&D, Irvine, CA), anti-PSCA antibody (1:50, #abx236848, Abbexa, Cambridge, UK).”
- 1(luminal cells) and Chromogranin A (neuroendocrine cells) should be included in figure 1D.
Thank you for the comment. Chromogranin A does not show expression in the stem/progenitor population and is not included in the heatmap. We used synaptophysin to represent the neuroendocrine cell population. We are not sure what is the other gene asked for luminal cells, as it only comes out as “1”.
- In the text, please mention about that SOX9 is also enriched in CFSEHigh cells in figure 1D.
We added in the text “Of note, transcription factor SOX9 was enriched in both label-retaining stem cells and non-retaining progenitor cells.” And stated in Figure Legend “SOX9 was uniquely enriched in CFSEHigh cells and a subpopulation of CFSELow cells.”
- In KRT5, 8 and 18 expressions of CFSELow cells, luminal and basal lineage does not look separated completely figure 1D. Please explain about this.
In the current study, we gradually teased out the cell lineage step-by-step and present the keratin genes profiling in different cell populations and then their subpopulations in separated figures in an order. The heatmap clustering in Figure 1D is the first step to separate the keratin genes in between CFSEHigh and CFSELow cells. Subclusters of prostate stem cells in CFSEHigh population were identified as quiescent and active stem cells as well as bipotent progenitors in Figures 2&3. While separation of basal and luminal progenitor cell lineages was further determined by heatmap and TSCAN in Figure 4.
For result 2.2 (Figure 2)
The authors defined cluster I as quiescent stem-like cells, cluster II as active stem cells, and cluster III as bipotent progenitor. However, the evidence for defining clusters is not mentioned clearly (line 147-149).
As mentioned above and in the manuscript, “In the current study, we gradually teased out the cell lineage step-by-step and present the keratin genes profiling in different cell populations and then their subpopulations in separated figures in an order”. For the label retaining cells that our 2017 and 2019 publications document possess all functional characteristics of stem cells (e.g. asymmetric cell division, tissue regeneration with low cell #, hallmark stem cell genes, etc) with performed RNA deep sequencing initially. This was followed by single cell analysis where bioinformatics identified 3 distinct clusters based on gene expression profiles. This evidence is provided in Figure 2 and 3. To understand what these 3 clusters represent and their relationships to eachother, we initially separated the CFSEHigh cells into three subpopulations by UMAP clustering (Fig 2A). Following this, single cell analysis using TSCAN, cell hierarchical clustering was reconstructed by minimum spanning tree (MST) incorporated with the traveling salesman problem algorithm to minimize the distance linking cell clusters. A pseudo-temporal ordering score was generated to determine starting and ending points of the tree (Fig 2B), which closely mimics the true biological time. Together, this identified cluster IàIIàIII as the biologic order wherein the three clusters arose. Based on these analyses and our previous studies that demonstrated the CFSEHigh cells as having stemness properties, Cluster I, the originating cells, are predicted to be quiescent stem-like cells, yielding cluster II, likely active stem cells that lead to cluster III, proposed as bipotent progenitor cells. Our prediction was supported by temporal tracing of three stemness genes PSCA, SPINK1 and CD36, which showed the enrichments of all three genes in cluster I with decreasing expression through cluster II and III. Further evidence shown in Fig 3A,B,C indicated the enrichment of stem cell related keratin genes in cluster I with progenitor cell related keratin genes in cluster II and III.
- Markers of active stem cells and bipotent progenitor are also required for cluster definition (Figure 2D).
We revised the text “Enrichment of KRT 13, 80, 78 and 23 was confirmed for Cluster I quiescent stem cells, while KRT16 (16P2, 16P4, 16P5) and 17 (17P2, 17P3, 17P6) were enriched in Clusters III bipotent progenitor cells. Active stem cells in cluster II express decreased levels of stemness keratin genes enriched in Clusters I and increased levels of KRT16 (16P2, 16P4, 16P5) and 17 (17P2, 17P3, 17P6) enriched in Clusters III”. It is important to point out here that these separate cell populations - with stemness activity - have not been previously interrogated in this manner, so this is the first report of putative quiescent and active stem cells and bipotent progenitor cells for the human prostate gland. As such, there are no known markers for these 3 populations to use for confirmatory studies. Enriched genes in these separate populations are currently being interrogated in depth in our lab and will be the subject of a future publication and the entire gene database for this study will be deposited and made public.
- In the reference article, SPINK1 is studied in prostate cancer cells. Therefore, if possible, other stemness markers in normal prostate stem cells are desirable.
We also show enrichment of PSCA and CD36 which were previously found in normal stem cells in the prostate and other organs.
For result 2.3(Figure 3)
- Please mentions about Cluster II in this part.
We added “Active stem cells in cluster II express decreased levels of stemness keratin genes enriched in Clusters I and increased levels of KRT16 (16P2, 16P4, 16P5) and 17 (17P2, 17P3, 17P6) enriched in Clusters III”.
For result 2.4(Figure 4)
- The authors should show the analyses with KRT5 and non-keratin markers analysis.
We focused on keratin profiling in prostate epithelial cell hierarchy and KRT5, a known prostate basal cell keratin, was clearly shown to be enriched in cluster IV and gradually decreased in cluster V and cluster VI in Figure 4A.
- Time course analysis with prostasphere should be performed to show cell differentiation.
Prostasphere cells at day 7 are not differentiated cells but rather are cells in the early lineage commitment. They contain a small number of prostate stem cells and a majority of early- and late-stage progenitor cells with basal or luminal lineage commitment. We separated CFSEHigh from CFSELow cells using sphere-based label-retaining assay followed by FACS to sort the stem-like cells from the progenitor cells (Hu et all, 2017, 2019), then ran Fluidigm single cell capture and RNA-seq separately for CFSEHigh and CFSELow cell populations. Three subpopulations each in CFSEHigh and CFSELow cells were identified by single cell clustering analysis. In each prostasphere, CFSEHigh cells are clearly derived from one single prostate stem cells that initiated the formation of the sphere which permits time-couse analysis as discussed above. However, for the CFSELow cells, the situation is different; the cells are derived from bipotent progenitors (proposed for Cluster III) that divide and yield two separated basal and luminal progenitor cell lineages. For pseudo-time reconstruction, the starting and ending points can only be accurately identified in clusters of cells with single cell origin such as CFSEHigh cells but not CFSELow cells with two cell origins. We therefore identified basal and luminal progenitor cell lineages in CFSELow cells using heatmap clustering. Relative KRT8 and KRT18 expression was used to identify early and late stages of luminal progenitor cells in the same lineage; however, not basal progenitor cells which are in the different lineage in this proposed progression sequence.
For result 2.5(Figure 5)
- In material and method, protocol of siRNA is not mentioned (related to figure 5E).
Thanks for the suggestion. We added the section below.
4.3. Gene knockdown by siRNA
For siRNA knockdown of KRT13 gene, PrE-Ca cancer cells (~70% confluence) in 2D culture were incubated with 40 nM of siControl (scramble) or siKRT13 (IDT, Coralville, IA) for 6 hours, respectively. 1X104/well of siRNA treated PrE-Ca cells with or without CFSE labeling were resuspended in 1:1 Matrigel/PrEGM plated in a 24-well plate for 7 days to form tumor sphere, medium was replenished with fresh 40 nM siRNA on day 3 of 3D culture. Tumor sphere numbers and the CFSEHigh% cells were evaluated as described previously [23,24].
- To confirm the inhibitory effect of KRT13 knockdown in stem cell self-renewal, please consider to inhibit KRT13 in CFSEHighcells of prostasphere and culture for 5 days to reconstruct prostaspheres.
We used primary culture cells from both normal and cancer specimen for the current study. The primary cells can be passaged for only few times and the CFSEHigh cells isolated from prostaspheres by FACS sorting will not survive well with two rounds of siRNA knockdown (as lipofectamine2000 is toxic to the primary cells) before moving into 2nd passage of 3D sphere formation. Therefore, we performed the siRNA knockdown on all sphere cells including CFSEHigh and CFSELow cells.
For result 2.6(Figure 6)
- In vivo analysis should be performed for confirmation of stem cell identity.
We thank reviewer for the comment. The in vivo analysis that confirm the stemness of the CFSEHigh cells was published in Hu et al, 2017 in a rigorous series of assays considered gold standards for stem cell identity. As the editor requests the submission of the revised version in 10 days, it is technically impossible to perform additional in vivo experiments which is also out of the scope of the current study.
For result 2.7 (Figure 7)
- The authors should show the analyses with KRT5 and non-keratin markers analysis.
We performed single cells RNA-seq followed by KRT gene profiling using heatmap (Figure 7C) and dotplot (Figure 7D) analyses to identify a group of keratin genes (KRT4, 13, 80, 78, 23, 10, 19, 16, 6A, 6B, 6C, 15, and 17) enriched in the cSC cluster. We showed the clustering of KRT13 plot as a representative marker for cancer stem-like cells. KRT5 is not enriched in cancer stem-like cells. Non keratin markers are not the focus in our study.
- Time course analysis with prostasphere should be performed to track cell differentiation.
Thank you for the comment. Please refer to our answer for 2.4
For discussion (Figure 8)
Difference between cluster IV and V was not fully mentioned. Therefore, differentiation from III to IV and V in the model is not clear (Time course analysis of prostasphere reconstruction from CFSEHigh might be useful for confirmation of the model).
We did clearly describe in the text at page 378-380 that “Cluster IV cells are unique potent basal progenitors enriched KRT5, 14, 6, 16 while cluster V cells are early-stage luminal progenitors enriched for KRT8, 18, 10”. For pseudo-time reconstruction question, please refer to our answer for 2.4.
- Keratin profiling of prostate cells is interesting for studying development/regeneration of prostate. However, Analysis of keratin profiling with in vitro cultured samples is not sufficient and with in vivo samples is also required.
This type of analysis in vivo would be extremely difficult to undertake, if not impossible. An advantage of the spheroid model is that without the typical stem cell growth restraints of the in vivo stem cell niche, the quiescent stem cells can divide and derive these populations in sufficient numbers to perform this detailed gene profiling analysis. In vivo, capturing these stages from the rare stem cells (which infrequently divide) at these sperate stages with proof that they come from an originated cell would not be possible. We could use enriched gene markers for the different proposed stages in combinations to sort these out using human tissue samples and these are in our plans for future studies. .
For some comments
- In line 265, “confirmed the enrichment of of specific”→ “confirmed the enrichment of specific”? Corrected in the revised version.
- In line 350, “shifts were foiund in the non-labeled progenitor” → “shifts were found in the non-labeled progenitor”? Corrected in the revised version.
For comments in reference
- 23: Hu WY, Hu DP, Xie L, Li Y, Nonn L, Shioda T, Prins GS. →Hu WY, Hu DP, Xie L, Li Y,Majumdar S, Nonn L, Shioda T, Prins GS. Corrected in the revised version.
- 59: 67, 4708-4715 → 67, 4807-4815 Corrected in the revised version.
Round 2
Reviewer 2 Report
- Between CFSEHigh and CFSELow, the brightness of red channel looks different in figure 1C.
The brightness of red channel between CFSEHigh and CFSELow is matched in the new Figure 1C.
To me, they have not been matched, yet. The background brightness is obviously different. And even from current new pictures, it is difficult to say the difference in KRT19 and KRT4 expression level between CFSE high and low populations.
- 1(luminal cells) and Chromogranin A (neuroendocrine cells) should be included in figure 1D.
Thank you for the comment. Chromogranin A does not show expression in the stem/progenitor population and is not included in the heatmap. We used synaptophysin to represent the neuroendocrine cell population. We are not sure what is the other gene asked for luminal cells, as it only comes out as “1”.
“1” was supposed to be “NKX3.1”.
- Markers of active stem cells and bipotent progenitor are also required for cluster definition (Figure 2D).
We revised the text “Enrichment of KRT 13, 80, 78 and 23 was confirmed for Cluster I quiescent stem cells, while KRT16 (16P2, 16P4, 16P5) and 17 (17P2, 17P3, 17P6) were enriched in Clusters III bipotent progenitor cells. Active stem cells in cluster II express decreased levels of stemness keratin genes enriched in Clusters I and increased levels of KRT16 (16P2, 16P4, 16P5) and 17 (17P2, 17P3, 17P6) enriched in Clusters III”. It is important to point out here that these separate cell populations - with stemness activity - have not been previously interrogated in this manner, so this is the first report of putative quiescent and active stem cells and bipotent progenitor cells for the human prostate gland. As such, there are no known markers for these 3 populations to use for confirmatory studies. Enriched genes in these separate populations are currently being interrogated in depth in our lab and will be the subject of a future publication and the entire gene database for this study will be deposited and made public.
Please describe above explanation about future studies in Discussion.
- To confirm the inhibitory effect of KRT13 knockdown in stem cell self-renewal, please consider to inhibit KRT13 in CFSEHighcells of prostasphere and culture for 5 days to reconstruct prostaspheres.
We used primary culture cells from both normal and cancer specimen for the current study. The primary cells can be passaged for only few times and the CFSEHigh cells isolated from prostaspheres by FACS sorting will not survive well with two rounds of siRNA knockdown (as lipofectamine2000 is toxic to the primary cells) before moving into 2nd passage of 3D sphere formation. Therefore, we performed the siRNA knockdown on all sphere cells including CFSEHigh and CFSELow cells.
Then, how is it possible to conclude that the loss of KRT13 likely inhibits stem cell self-renewal (line 261-262)? Are other KRTs expressed in CFSE low not important for the spheroid formation? Does KRT13 (as a KRT in CFSE high) siRNA inhibit spheroids formation more efficiently than siRNAs of other KRTs in CFSE low? With this only experiment in fig.5E, it is difficult to conclude the role of KRT13 in these cancer stem cells.
- Keratin profiling of prostate cells is interesting for studying development/regeneration of prostate. However, Analysis of keratin profiling with in vitro cultured samples is not sufficient and with in vivo samples is also required.
This type of analysis in vivo would be extremely difficult to undertake, if not impossible. An advantage of the spheroid model is that without the typical stem cell growth restraints of the in vivo stem cell niche, the quiescent stem cells can divide and derive these populations in sufficient numbers to perform this detailed gene profiling analysis. In vivo, capturing these stages from the rare stem cells (which infrequently divide) at these sperate stages with proof that they come from an originated cell would not be possible. We could use enriched gene markers for the different proposed stages in combinations to sort these out using human tissue samples and these are in our plans for future studies. .
Then, please be advised to add a panel like fig.1A into fig.5A to show all specimen were obtained from in vitro spheroid cultures. I know the description is in line 231-232, but I think it would be helpful for readers to understand the experiment plan at a glance.
Author Response
- Between CFSEHigh and CFSELow, the brightness of red channel looks different in figure 1C.
The brightness of red channel between CFSEHigh and CFSELow is matched in the new Figure 1C.
To me, they have not been matched, yet. The background brightness is obviously different. And even from current new pictures, it is difficult to say the difference in KRT19 and KRT4 expression level between CFSE high and low populations.
Thanks for the comment. Reviewer should be aware that we are comparing prostate stem cells with their daughter progenitor cells but not fully differentiated basal or luminal epithelial cells. Therefore, in term of stemness markers, we should not expect a huge difference in either mRNA or protein expression. This panel of stemness keratin genes enriched in CFSEHigh cells were only recently discovered by our single cell RNA-seq analysis, most of them did not show up in our previous bulk RNA seq analysis (Hu et al, 2017, KRT13 was identified as stemness keratin). We used ICC staining to supports the enrichments of these keratins at protein level. The images clearly showed the increased protein amount and light intensity of KRT80, KRT78, KRT23, KRT19 and KRT4 in BrdU-retaining or CFSEHigh cells regardless the slightly difference in background brightness, this is also consistent with our previous reports (Hu et al, 2017, 2019) showing that CFSEHigh cells are larger in size. If the reviewer wants to argue the small difference in background, the red channel in CFSELow cells is actually a little bit brighter while the keratin proteins are more abundant in CFSEHigh cells.
- 1(luminal cells) and Chromogranin A (neuroendocrine cells) should be included in figure 1D.
Thank you for the comment. Chromogranin A does not show expression in the stem/progenitor population and is not included in the heatmap. We used synaptophysin to represent the neuroendocrine cell population. We are not sure what is the other gene asked for luminal cells, as it only comes out as “1”.
“1” was supposed to be “NKX3.1”.
Thanks for the suggestion. We have shown the expressions of differentiation markers including NKX3.1 and E-cadherin progenitor cells in our previous publications (Hu et al, 2011, 2017, 2019). The current study focuses on keratin profiling on prostate stem/progenitor cells, therefore, we properly used KRT8 and KRT18 to represent the luminal cell cluster.
- Markers of active stem cells and bipotent progenitor are also required for cluster definition (Figure 2D).
We revised the text “Enrichment of KRT 13, 80, 78 and 23 was confirmed for Cluster I quiescent stem cells, while KRT16 (16P2, 16P4, 16P5) and 17 (17P2, 17P3, 17P6) were enriched in Clusters III bipotent progenitor cells. Active stem cells in cluster II express decreased levels of stemness keratin genes enriched in Clusters I and increased levels of KRT16 (16P2, 16P4, 16P5) and 17 (17P2, 17P3, 17P6) enriched in Clusters III”. It is important to point out here that these separate cell populations - with stemness activity - have not been previously interrogated in this manner, so this is the first report of putative quiescent and active stem cells and bipotent progenitor cells for the human prostate gland. As such, there are no known markers for these 3 populations to use for confirmatory studies. Enriched genes in these separate populations are currently being interrogated in depth in our lab and will be the subject of a future publication and the entire gene database for this study will be deposited and made public.
Please describe above explanation about future studies in Discussion.
Thanks for the great comment, we added this in the Discussion section lines 372-378 “It is important to point out here that these separated cell populations - with stemness activity - have not been previously interrogated in this manner, so this is the first report of putative quiescent and active stem cells and bipotent progenitor cells for the human prostate gland. As such, there are no known markers for these 3 populations to use for confirmatory studies. Enriched genes in these separate populations are currently being interrogated in depth in our lab and will be the subject of a future publication”.
- To confirm the inhibitory effect of KRT13 knockdown in stem cell self-renewal, please consider to inhibit KRT13 in CFSEHighcells of prostasphere and culture for 5 days to reconstruct prostaspheres.
We used primary culture cells from both normal and cancer specimen for the current study. The primary cells can be passaged for only few times and the CFSEHigh cells isolated from prostaspheres by FACS sorting will not survive well with two rounds of siRNA knockdown (as lipofectamine2000 is toxic to the primary cells) before moving into 2nd passage of 3D sphere formation. Therefore, we performed the siRNA knockdown on all sphere cells including CFSEHigh and CFSELow cells.
Then, how is it possible to conclude that the loss of KRT13 likely inhibits stem cell self-renewal (line 261-262)? Are other KRTs expressed in CFSE low not important for the spheroid formation? Does KRT13 (as a KRT in CFSE high) siRNA inhibit spheroids formation more efficiently than siRNAs of other KRTs in CFSE low? With this only experiment in fig.5E, it is difficult to conclude the role of KRT13 in these cancer stem cells.
Thank reviewer for the insightful comment. We have documented KRT13 as stemness keratin gene in prostate stem cells in our previous publication (Hu et. al., 2017, 2019). We used siRNA to knockdown of KRT13 in normal prostate epithelial cells and found decreased number of prostasphere formation as well as decreased label-retaining prostate stem cells, both are readouts of stem cell self-renewal. In the current study, we knockdown KRT13 in prostate cancer cells and found the similar results with decreases of both tumor sphere formation and label-retaining cancer-stem like cells, these are sufficient in suggesting the critical role of KRT13 in cancer-stem like cells.
In addition to the findings of KRT13 from our lab, several other research groups including Dr. Leland WK Chung, Dr. Isla P Garraway, and Dr. Douglas W Strand, all are experts in the prostate stem cell field, have documented KRT13 as an important stem cell keratin in both prostate stem cells and cancer stem-like cells.
- Keratin profiling of prostate cells is interesting for studying development/regeneration of prostate. However, Analysis of keratin profiling with in vitro cultured samples is not sufficient and with in vivo samples is also required.
This type of analysis in vivo would be extremely difficult to undertake, if not impossible. An advantage of the spheroid model is that without the typical stem cell growth restraints of the in vivo stem cell niche, the quiescent stem cells can divide and derive these populations in sufficient numbers to perform this detailed gene profiling analysis. In vivo, capturing these stages from the rare stem cells (which infrequently divide) at these sperate stages with proof that they come from an originated cell would not be possible. We could use enriched gene markers for the different proposed stages in combinations to sort these out using human tissue samples and these are in our plans for future studies.
Then, please be advised to add a panel like fig.1A into fig.5A to show all specimen were obtained from in vitro spheroid cultures. I know the description is in line 231-232, but I think it would be helpful for readers to understand the experiment plan at a glance.
Yes, in agree with the reviewer, we did describe the cell source in detail for each experiment at the beginning, please see lines 102-107 “To investigate the keratin gene profiles in prostate stem and progenitor cells, we pre-labeled pooled primary prostate epithelial cells (PrECs) from three normal prostate tissues with BrdU or 5(6)-carboxyfluorescein N-hydroxysuccinimidyl ester (CFSE) followed by wash-out in 3D Matrigel culture for 5 days during spheroid formation. Prostaspheres were dispersed into single cells and CFSEHigh prostate stem-like cells (1.01%) were separated from CFSELow progenitor cells (97.52%) by FACS (Fig. 1A) [23,24]”.